# LEARNING TO MAKE MISTAKES: MODELING INCORRECT STUDENT THINKING AND KEY ERRORS

## ABSTRACT

Research on reasoning in language models (LMs) predominantly focuses on improving the correctness of their outputs. But some important applications require modeling reasoning patterns that are *incorrect*. For example, automated systems that can reason about and simulate student errors are useful for providing real-time feedback in the classroom or offline practice for educators-in-training. This paper presents a new method, MISTAKE, that (1) constructs high-quality synthetic examples of reasoning errors by leveraging cycle consistency between incorrect answers and latent misconceptions; and (2) uses the generated data to learn models for student simulation, misconception classification, and answer generation. We evaluate MISTAKE on three educational tasks and find that it results in (1) higher accuracy when *simulating incorrect student answers* based on specific misconceptions, (2) increased performance *inferring latent misconceptions* from observed incorrect answers, and (3) higher alignment with expert-written distractor answers when *generating incorrect answers* (*e.g.,* for multiple-choice tests).

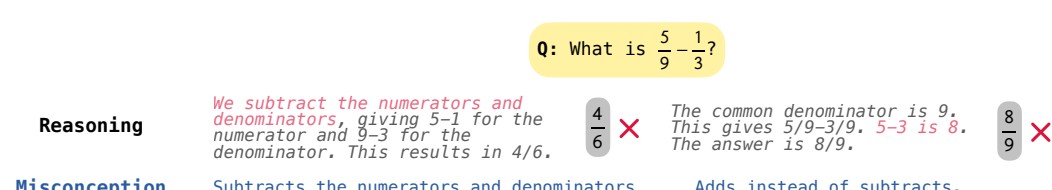

Figure 1: Examples of mathematical errors that result from common misconceptions shared among students.

## 1 INTRODUCTION

There is a substantial body of language model (LM) research focused on generating high-quality reasoning traces that lead to correct answers (Wei et al., 2022; Nye et al., 2022; Zelikman et al., 2022). However, many applications of LMs require modeling how reasoning can be *wrong*. For example, in education, being able to understand the common reasoning errors that students make allows for tailored assessment and instruction. In addition, recent work has applied LMs to simulate students for uses such as teacher training (Markel et al., 2023) and evaluating AI tutors (Wang et al., 2025; Liu et al., 2024), both of which require being able to simulate their incorrect reasoning. Outside of education, work in the social sciences on simulating human behavior with LMs, for example in psychology (Dillion et al., 2023; Demszky et al., 2023; Park et al., 2024) and economics (Filippas et al., 2024), also requires being able to model cognitive biases and fallacies.

Figure 1 shows exemplary examples of common incorrect reasoning exhibited by students in an elementary mathematics setting. The figure gives examples of two errors in solving a question about fractions; these particular errors result from specific misconceptions shared by many learners encountering fraction arithmetic for the first time. Modeling such errors requires a nuanced understanding of the relationship between mathematical concepts and how people reason about them. As we show, current LMs are much worse at simulating such errors than they are at performing correct reasoning to, *e.g.,* solve math problems.

In this paper, we introduce a *self-supervised* procedure for generating high-quality reasoning data that models the underlying patterns in student errors, such as those shown in Figure 1. The key idea behind our approach is to leverage cycle consistency between incorrect answers and their underlying misconceptions; this allows us to augment a set of questions with misconceptions, reasoning, and incorrect answers without requiring any examples of human-generated errors. We then use this data to improve performance on three education tasks. We refer to the end-to-end method as MISTAKE (MODELING INCORRECT STUDENT THINKING AND KEY ERRORS).[1]

MISTAKE is built from two procedures. The **inner loop**, MISTAKE-GENERATE, samples plausible triples (misconception, faulty reasoning, answer) by decoding from a model with a cycle consistency constraint. The **outer loop**, MISTAKE-UPDATE, fine-tunes models on the cycle consistent data. Together, they provide an end-to-end, self-supervised procedure for generating large numbers of synthetic reasoning traces with interpretable errors; they additionally yield both a **student simulation** model capable of simulating *reasoning with misconceptions*, and a **misconception inference** model that can observe a student's behavior and *reason about misconceptions* to identify what the student is confused about.

Models trained via MISTAKE achieve improved performance on three education tasks that are directly useful for real-world applications in education:

1. **Student Simulation:** There has been a growing interest in simulating students, and more broadly users, with LMs in order to facilitate real-world evaluations of AI systems when access to real students (Macina et al., 2023; Wu et al., 2025b; Miroyan et al., 2025; Perczel et al., 2025) or users (Park et al., 2024; Wu et al., 2025a; Naous et al., 2025) is not available. A key requirement for useful student simulators is being able to *simulate their mistakes*. Given a misconception, we evaluate how well an LM can simulate the incorrect reasoning and answer that a student would produce. MISTAKE improves accuracy by up to **9%** (§5.2).

2. **Misconception Inference:** Building personalized educational systems such as LLM-based tutors that can adapt to individual students requires being able to make inferences about students' misconceptions (Ross & Andreas, 2024). This task involves inferring a student's misconception based on an incorrect answer they provided. MISTAKE leads to a **15%** improvement in performance on this task (§5.3).

3. **Distractor Generation:** Methods for automatic generation of distractors for multiple-choice problems are used to generate high-quality assessment problems for students (McNichols et al., 2024; Feng et al., 2024). This task evaluates MISTAKE's ability to generate high-quality incorrect distractor answers. MISTAKE generates distractor answers that are more often found in the expert-written distractor choices for each question, with a **64.6%** increase in precision, suggesting that MISTAKE generates incorrect data that is more aligned with the kinds of mistakes that students make (§5.4).

Together, our results highlight the promise of explicitly modeling patterns of incorrect reasoning across a range of educational domains.[2]

## 2 RELATED WORK

**Education** Work on modeling student misconceptions has a long history in education research (Brown & Burton, 1978; van, 1990; Feldman et al., 2018), and more recently within AI for education. In a synthetic evaluation framework, Ross & Andreas (2024) find that LLMs can infer student misconceptions and adapt teaching strategies better than simple baselines but worse than more sophisticated methods that explicitly model misconceptions. Similarly, Scarlatos et al. (2025) find that combining LMs with knowledge tracing (KT) leads to better estimates of student knowledge states than KT-only methods in dialogue settings. Sonkar et al. (2024b) find that LLMs are much worse at

---

[1]We note that we do not aim to generate reasoning traces or rationales that are themselves human-like, but instead our goal is to develop models that can better model the underlying *patterns* in student errors. Improved performance at the student simulation and misconception inference tasks is direct evidence that models have learned to model the missteps in student reasoning traces, whether or not the form of the rationales themselves look like those that would be generated by human students.

[2]Our code is publicly available at URL.

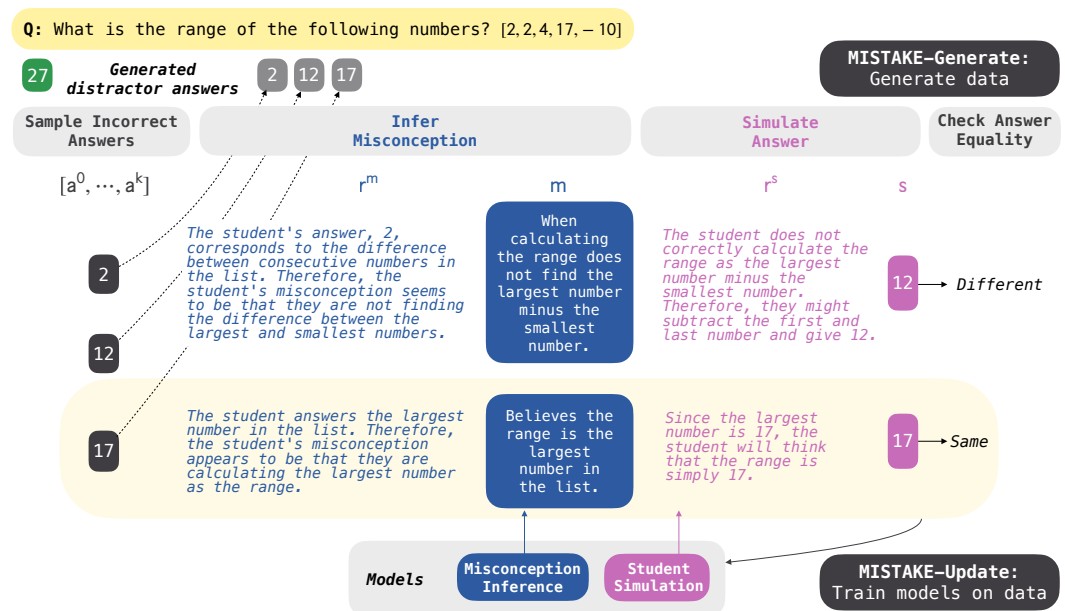

Figure 2: Overview of MISTAKE. MISTAKE-GENERATE generates data by enforcing cycle consistency between misconceptions, reasoning traces, and answers. MISTAKE-UPDATE iteratively trains student simulation and misconception inference models on this data, generates new data using MISTAKE-GENERATE and these models, and repeats.

identifying incorrect reasoning containing misconceptions than they are at identifying correct reasoning. All of these studies suggest that there is headway to be made in using LMs to explicitly model student misconceptions.

A key challenge in such research is the scarcity of high-quality data, particularly expert-annotated examples of real student misconceptions. The DrawEduMath dataset contains students' handwritten solutions annotated by expert teachers (Baral et al., 2024); however, while it contains annotations of students' errors and strategies used to solve the problem, it lacks standardized annotations of higher-level misconceptions; similarly, the MalAlgoQA dataset contains math problems with associated incorrect answers and incorrect rationales, but the incorrect rationales are again problem-specific (Sonkar et al., 2024b).[3] The EEDI Mining Misconceptions in Mathematics dataset (King et al., 2024) is one of a few datasets that contain natural student data with annotations of generalizable error descriptions. However, the process of collecting expert teacher annotations remains resource-intensive, limiting the scalability of these datasets.

In light of these data limitations, recent works have used off-the-shelf LMs to simulate students. Recent tutoring benchmarks use LM-simulated students for both dataset construction and evaluation (Macina et al., 2023; Daheim et al., 2024; Liu et al., 2024; Wang et al., 2025). Existing approaches predominantly aim to simulate general student performance or skills rather than specific misconceptions (Lu & Wang, 2024; Benedetto et al., 2024). While Sonkar et al. (2024a) propose a Python library that models misconceptions in linear algebra, their approach, based on a hand-engineered graphical model, is limited to specific types of equations. In contrast to this past work, MISTAKE provides a self-supervised method for generating high-quality data with misconceptions and learning models from this data that can simulate misconceptions in a natural educational domain.

Outside of student simulation, another promising educational application of AI is in helping automate *assessment*, *e.g.,* by constructing high-quality distractor answers for multiple-choice questions. Previous work has leveraged in-context learning with nearest-neighbor examples (McNichols et al., 2024; Feng et al., 2024). Scarlatos et al. (2024) introduce a ranking model to predict student se-

---

[3]For example, an incorrect rationale in the MalAlgoQA dataset is: "Chose the number of times a star is picked in the 1st 50 cards drawn." This is an incorrect reasoning step specific to a particular problem, not reflective of the kinds of higher-level misconceptions that affect student reasoning across math problems.

lection probabilities for distractors, using this to filter LM generated options, and Fernandez et al. (2024) introduce a method that jointly learns textual descriptions of the errors behind incorrect answers along with the incorrect answers. However, all of these methods require a dataset of existing distractors to use as candidates/training examples. As we will see, MISTAKE produces high-quality distractors as a byproduct of training, *without* a dataset of existing human-authored distractors.

**Reasoning**    Our work is also related to the literature on learning to reason (Wei et al., 2022; Nye et al., 2022; Li et al., 2023; Zelikman et al., 2022; 2024; DeepSeek-AI et al., 2025). Most closely related is STAR, an algorithm that iteratively samples reasoning traces from a model, trains on a filtered set of traces, re-samples, and repeats (Zelikman et al., 2022). Many follow up methods involve training external reward models, which are typically trained on human annotations (Ouyang et al., 2022; Dong et al., 2023). Unlike these works, MISTAKE is self-supervised and learns to impute both reasoning and target (incorrect) labels without annotations of either, using cycle consistency to filter out low-quality generations. Also related are self-supervised methods that use self-consistency to select an answer that is consistent across multiple reasoning paths (Wang et al., 2023) or use LMs as judges (Yuan et al., 2024) to evaluate generations. A key difference between MISTAKE and these existing self-supervised works is that MISTAKE involves training both a forward reasoning model (inferring an answer from a latent misconception) and an inverse reasoning model (inferring the latent reasoning pattern, i.e. misconception, from the answer), which as we show outperforms training just one of these models and keeping the other fixed.

# 3 MISTAKE (MODELING INCORRECT STUDENT THINKING AND KEY ERRORS)

Our ultimate goal is to train two distinct models: first a **student simulation model** $M_s$ that can generate plausible student behavior *conditioned* on student descriptions (which may include misconceptions); second a **misconception inference model** $M_m$ that can observe a student trace and likely sources of student errors. MISTAKE trains these models via two nested procedures: an inner loop MISTAKE-GENERATE (§3.1) that generates data by enforcing cycle consistency between inferred misconceptions, generated reasoning traces, and answers; and an outer loop MISTAKE-UPDATE (§3.2) that uses the data to finetune $M_s$ and $M_m$. Figure 2 shows an overview of MISTAKE with examples.

## 3.1 MISTAKE-GENERATE: SELF-SUPERVISED DATA GENERATION

Algorithm 1 presents an overview of MISTAKE-GENERATE, which uses an existing base LM $M$, student model $M_s$, and misconception model $M_m$ to generate new traces exhibiting reasoning with misconceptions. Below we explain how the procedure works step-by-step.

---

**Algorithm 1** MISTAKE-GENERATE: Self-Supervised Data Generation

**Input:** Questions $Q$, pretrained model $M$, student simulation model $M_s$, misconception inference model $M_m$

1: **for** each question and correct answer pair $(q, a^*) \in Q$ **do**
2:     $[a_0, a_1, a_2] \leftarrow \texttt{Sample\_Answers}(q, a^*, M)$ # Sample 3 incorrect answers with $M$
3:     $q_{mc} \leftarrow (q, a_0, a_1, a_2, a^*)$ # Create a multiple choice question
4:     **for** each incorrect answer $a$ **do**
5:         $r^m, m \leftarrow \texttt{Infer\_Misconception}(q_{mc}, a, M_m)$ # Infer misconception with $M_m$
6:         $r^s, s \leftarrow \texttt{Simulate\_Student}(q_{mc}, m, M_s)$ # Simulate student based on $m$ with $M_s$
7:         $w \leftarrow \begin{cases} \alpha & \text{if } \texttt{Check\_Cycle}(a, s, a^*, M) \\ 1 & \text{otherwise} \end{cases}$ # Check cycle consistency with $M$
8:         Add $(q_{mc}, r^s, s, r^m, m, w)$ to dataset $D$
9:     **end for**
10: **end for**
11: **return** Dataset $D$ of weighted examples

---

**Sample_Answers**  The first step in MISTAKE-GENERATE is to sample a set of incorrect answers $[a_0, \cdots, a_k]$ that a student might have when solving a question $q$. We sample these answers by prompting a pretrained LM $M$, conditioning on the question $q$ and the correct answer $a^*$. The generated answers are used as (a) distractors for the student simulation module Simulate_Student, which takes in multiple-choice questions, and (b) as candidate labels for the misconception inference module Infer_Misconception module and rest of the MISTAKE-GENERATE process. For example, for the question shown in Figure 2, [*What is the range of the following numbers? [2, 2, 4, 17, -10]*], Sample_Answers may output $[2, 12, 17]$.

**Infer_Misconception**  Given the multiple choice question $q_{mc}$ with generated distractor answers and specific candidate answer $a$, the Infer_Misconception module uses the misconception model $M_m$ to infer the conceptual misunderstanding that would have led to the incorrect answer $a$. The outputs of Infer_Misconception are the inferred misconception $m$, along with a reasoning trace $r^m$ explaining how it arrived at that conclusion. For example, for candidate answer $a = 17$, Infer_Misconception may output $r^m$ = [*The student answers the largest number in the list. Therefore, the student's misconception appears to be that they are calculating the largest number as the range*] and $m$ = [*Believes the range is the largest number in the list*].

**Simulate_Student**  Given a question $q_{mc}$ and inferred misconception $m$, Simulate_Student uses the student simulator $M_s$ to simulate the step-by-step reasoning and final answer that a student would produce if they had the misconception. For example, for misconception $m$ = [*Believes the range is the largest number in the list*], Simulate_Student may output $r^s$ = [*The student answers the largest number in the list. Therefore, the student's misconception appears to be that they are calculating the largest number as the range*] and $s$ = 17.

**Check_Cycle**  The cycle consistency check serves as a self-supervised quality filter. If Check_Cycle returns true, this provides strong evidence that the inferred misconception $m$ has the desired relationship with the original answer $a$. This is because if the misconception were incorrect or unrelated to the answer it would be unlikely that simulating a student with that misconception would produce the same answer again. For example, the first misconception in Figure 2, [*Believes the range is the largest number in the list*], is a high-quality misconception and, when simulated faithfully, should lead to the original answer $a = 17$. The cycle consistency check therefore verifies both directions of the relationship: that the misconception explains the original answer (answer $\rightarrow$ misconception) and that the misconception leads back to the same answer (misconception $\rightarrow$ answer). Examples that pass this check are given higher weight ($w = \alpha$) in the training data, as they represent more reliable examples of the relationship between misconceptions and incorrect answers.

There are some boundary cases for the cycle consistency check. For example, the second misconception [*When calculating the range does not find the largest number minus the smallest number*] is too general to be able to re-simulate the exact original sampled answer $s = 2$, as it could explain many incorrect answers. However, we may still want to include the re-simulation [*The student does not... Therefore, they might subtract the first and last number and give 12*] since it may still be useful for learning how to generally simulate student mistakes, as long as it leads to an incorrect answer. For this reason, we explore two variants of MISTAKE (§4.3): one that filters misconceptions based on the *strong* constraint that the inferred misconception results in the same incorrect answer that was sampled (*i.e.,* $s = a$), which we call MISTAKE-CYCLE+CORRECT, and another that uses the *weaker* constraint that the simulated answer is not the correct answer (*i.e.,* $s \neq a^*$), which we call MISTAKE-CYCLE.

### 3.2 MISTAKE-UPDATE: ITERATIVE TRAINING ALGORITHM

MISTAKE-UPDATE is an iterative algorithm that trains two models on related tasks using the data generated by MISTAKE-GENERATE as described in §3.1. Algorithm 2 summarizes the iterative training process used to train the student simulation model $M_s$ and the misconception inference model $M_m$.

We subset the data generated by MISTAKE into two datasets: one for training a student simulation model $M_s$ and one for training a misconception inference model $M_m$. $M_s$ is trained on the simulated

incorrect answers $s$ and reasoning traces $r^s$ used to generate those answers, while $M_m$ is trained on the incorrect answers $s$ and inferred misconceptions $m$.

---

**Algorithm 2** MISTAKE-UPDATE: Iterative Training of Student Simulation and Misconception Inference Models

---

**Input:** a pretrained language model $M$

1: $D_0 \leftarrow$ MISTAKE$(M, M)$ # Generate initial dataset with MISTAKE using $M$
2: $D_0^s \leftarrow \{(x = (q, m), y = (r^s, s)) \mid (q, r^s, s, r^m, m) \in D_0\}$ # Student simulation data
3: $D_0^m \leftarrow \{(x = (q, s), y = (r^m, m)) \mid (q, r^s, s, r^m, m) \in D_0\}$ # Misc. inference data
4: **for** $t = 1$ to $T$ **do**
5: $\quad M_s \leftarrow$ train$(M, D_{t-1}^s)$ # Finetune orig model on new student simulation data
6: $\quad M_m \leftarrow$ train$(M, D_{t-1}^m)$ # Finetune orig model on new misconception inference data
7: $\quad D_t \leftarrow$ MISTAKE$(M_s, M_m)$ # Generate new MISTAKE data with finetuned $M_s, M_m$
8: $\quad D_t^s \leftarrow \{(x = (q, m), y = (r^s, s)) \mid (q, r^s, s, r^m, m) \in D_t\}$ # Student simulation data
9: $\quad D_t^m \leftarrow \{(x = (q, s), y = (r^m, m)) \mid (q, r^s, s, r^m, m) \in D_t\}$ # Misc. inference data
10: **end for**
11: **return** $M_s, M_m$ # Return trained models

---

Inspired by STAR (Zelikman et al., 2022) and other expectation-maximization-style algorithms for training LMs (*e.g.,* Bostrom et al., 2024), we iteratively finetune $M_s$ and $M_m$ on the data generated by MISTAKE-GENERATE, using the finetuned models to generate new data, and repeating. MISTAKE-UPDATE seeds the iterative process by using a pretrained LM $M$ as $M_s$ and $M_m$ to generate the initial dataset $D_0$. After the first iteration, the finetuned models are used to generate the next round of data with MISTAKE-GENERATE, which is used to finetune the models again. This process repeats for $T$ iterations. The final results are trained $M_s$ and $M_m$ models useful for simulating student reasoning and inferring misconceptions respectively. Importantly, both $M_s$ and $M_m$ are reasoning models—in contrast to existing EM-style training procedures for LMs, both the inference model and the forward simulation model "think out loud" and improve their behavior over time.

## 4 EXPERIMENTS

In this section, we describe our experiments evaluating MISTAKE on three education tasks.

### 4.1 DATA

We work with the EEDI Mining Misconceptions in Mathematics dataset, which consists of 1,857 K–12 math questions (King et al., 2024). Each question has four expert-written multiple choice options that correspond to misconceptions that a student might have.[4] The incorrect answer choices and misconception annotations in EEDI are written by expert educators. We evaluate on these labels to determine whether MISTAKE, which only ever trains models on synthetically generated misconception data, generalizes to *real-world* data.

We subset the EEDI data into train (70%), validation (15%), and test splits (15%) by holding out math questions so that all (question, misconception, answer) pairs for the same question end up in the same split. We report results on the test set unless otherwise specified.

### 4.2 TASKS

We evaluate MISTAKE on three tasks that are useful for tailoring assessment and instruction to different students and providing offline practice for educators-in-training.

**Student Simulation** We evaluate a model's ability to simulate the incorrect answer that a student with a particular misconception would give. For each incorrect multiple choice answer in EEDI that has a labeled misconception, we evaluate whether the incorrect answer generated by the student

---

[4] Of the 7,428 total answer choices in the dataset, 4,338 of them are labeled with text descriptions of corresponding misconceptions. There are 2,587 unique misconceptions in the dataset.

simulation model, conditioned on a misconception description, is the same as the ground truth incorrect answer corresponding to the misconception. We evaluate the **accuracy** of simulated answers through pattern matching on generated letters corresponding to answer choices.

**Misconception Inference**   We also run the evaluation in the reverse direction: We evaluate the misconception inference model's accuracy at predicting a student's latent misconception from the incorrect answer they gave. Given a math question, an incorrect multiple choice answer, and a ground-truth misconception associated with the incorrect answer, we prompt the misconception inference model to output a description of the misconception that would lead to the answer. To evaluate the generated misconception, we embed the generated misconception, ground truth misconception, and full list of possible misconceptions in the EEDI data. We use the `Instructor-XL` model to embed misconceptions (Su et al., 2023).[5] We then sort the list of candidate misconceptions by their cosine similarity to the generated misconception and evaluate the mean average precision at k, or **MAP@k** score, a metric introduced in the challenge along with the EEDI data:

$$\text{MAP@k} = \begin{cases} \frac{1}{p} & \text{if true misconception found at} \\ & \text{position p in top k misconceptions} \\ 0 & \text{otherwise} \end{cases}$$

where $p$ is the position where we find the true misconception in our sorted list of predictions. For example, if the true misconception appears at position 3 in our sorted list, then the score would be $\frac{1}{3}$. If the true misconception is not found in the top k predictions, the score is 0. We report results for k=25, as this is the value used by the EEDI Mining Misconceptions in Mathematics Challenge.[6]

**Distractor Generation**   We evaluate the ability of MISTAKE to generate human-aligned distractor answers. We measure the **precision** of generated distractor answers that match expert-written incorrect answers after filtering for cycle-consistency. For each (generated distractor, ground-truth distractor answer) pair, we prompt a judge LM (`GPT-4o-mini`) to determine whether they are equal (see Table 3 for the prompt). In a manual analysis of the `GPT-4o-mini` judge's annotations, we found that they were 100% accurate.[7] We then compute the proportion of distractor answers that are judged to be the same as at least one of the ground truth incorrect answers for the question.

### 4.3 METHOD VARIANTS

We experiment with several variants of MISTAKE that differ in `Check_Cycle` conditions. Table 7 summarizes the different variants. The first is **MISTAKE-CYCLE+CORRECT**, which uses the full cycle consistency criterion. In particular, MISTAKE-CYCLE+CORRECT *upweights* examples where the generated answer is fully cycle consistent—*i.e.,* the same as the answer sampled with `Sample_Answers` (*i.e.,* $s = a$)—and *removes* examples where the generated answer equals the correct answer, *i.e.,* $s = a^*$.[8] The second variant is **MISTAKE-CORRECT**, which only removes examples where the generated answer equals the correct answer, *i.e.,* $s = a^*$. The last variant is **NO-CYCLE**, which ablates both types of cycle consistency conditions and weights all examples equally.

We also ablate the joint training of student simulation and misconception inference models by only training one of the two models, holding the other fixed. We refer to these ablations as **STUDENT-ONLY** and **MISCONCEPTION-ONLY**.

---

[5]The instruction for the `Instructor-XL` embedding model is: [*Represent the following misconception that a student might have in solving K-12 math problems for retrieving similar misconceptions.*]

[6]The challenge can be found at: https://www.kaggle.com/competitions/eedi-mining-misconceptions-in-mathematics

[7]We validate the accuracy of the `GPT-4o-mini` judge by manually annotating 40 randomly sampled judgments of whether a generated distracted answer choice is the same as a ground truth answer choice. We find that all 40 answer judgments are correct. This high accuracy is explained by this judgment task being easy: The model simply needs to judge whether two answers are the same answer in different forms (*e.g.,* recognizing that the answer "Neither Tom nor Katie are correct" is the same as the answer "Neither is correct"), and therefore the `GPT-4o-mini` model can suffice for this task.

[8]We experimented with removing all examples that were not cycle consistent rather than upweighting ones that were, but found that this led to slightly worse results.

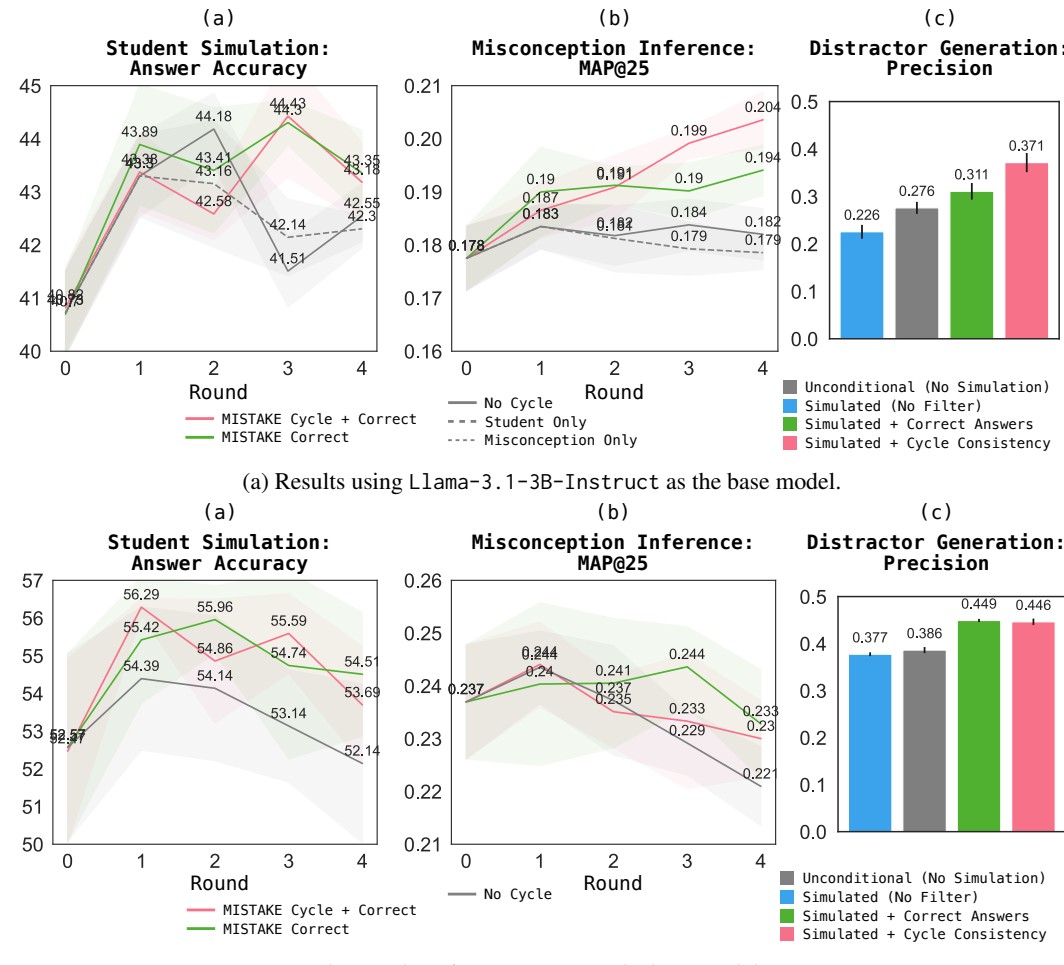

(a) Results using `Llama-3.1-3B-Instruct` as the base model.

(b) Results using `Qwen3-8B` as the base model.

Figure 3: Results on the three educational tasks described in §4.2. We report means and standard errors across 5 random seeds. (a) Student simulation accuracies of MISTAKE variants (§5.2) (test set). (b) Misconception inference results for MISTAKE variants (test set) (§5.3). (c) Precision of generated distractor answers for MISTAKE-CYCLE+CORRECT (validation set) (§5.4).

## 4.4 EXPERIMENTAL SET-UP

We experiment with two base models in our experiments: `Llama-3.1-8B-Instruct` (Grattafiori et al., 2024) and `Qwen3-8B` (Yang et al., 2025). We use the same model for all five steps in MISTAKE and in MISTAKE-UPDATE. We prompt all models with few-shot examples with manually written reasoning traces. See the Appendix for details. We run 5 random seeds per experiment.

In addition to the self-supervised quality filters described in §4.3, we filter examples where the generated data consists of empty strings, which happens if the model does not generate an output in the correct format.[9]

For MISTAKE-UPDATE, we fine-tune models using LoRA (Hu et al., 2022) with rank $r = 8$ for up to 4 epochs, with early stopping based on validation loss on the synthetically generated validation dataset. We run experiments for $T = 4$ iterations.[10]

---

[9]We remove examples where $r^s$ or $s$ are empty strings from $D^s$, and we remove examples where $r^m$ or $m$ are empty strings from $D^m$.

[10]We train all models on a single H100 GPU.

## 5 RESULTS

Figure 3 shows how MISTAKE variants and ablations perform across training rounds. We provide more detailed presentations of results for each task in the rest of the section. Tables 8 and 9 contain examples of model outputs for the student simulation and misconception inference tasks, respectively.

### 5.1 API MODEL REFERENCES

Tables 1 and 2 show how the best results achieved by a MISTAKE variant compares to prompting closed GPT models. We note that these prompted methods are not baselines in that MISTAKE could be applied on top of any existing model (as long as it is open); however, they are useful reference points for how frontier LMs perform on these tasks. Overall, we find that for student simulation and misconception inference, the best performing Llama-3.1-8B-Instruct models trained with MIS-TAKE perform comparably or better than GPT-3.5-turbo for student simulation and misconception inference, and approach the performance of models several orders of magnitude larger.

Because the cycle-consistency filtering procedure in MISTAKE-GENERATE can be applied before fine-tuning, we can also apply it directly to API models. Here we find that MISTAKE improves the precision of generated distractor across scales, including GPT-4o and GPT-4.1 models.

### 5.2 STUDENT SIMULATION

We find that all models achieve much lower accuracy on student simulation than for the task itself (solving the math questions correctly); the drop in accuracy ranges from **24.6%** (92.4% → 66.3%) to **45.2%** (74.1% → 40.6%). Even powerful LMs such as GPT-4o and GPT-4.1 struggle to simulate incorrect student answers. The pretrained Llama-3.1-8B-Instruct model performs poorly on the student simulation task, with a starting accuracy of **40.83%**, which is **58.8%** of the model's performance at the task of solving math problems. This difference suggests that student simulation is a more difficult task for current models than solving math correctly.

As shown in Figure 3a, we find that all MISTAKE variants lead to some accuracy improvements, but the methods with some version of cycle consistency—MISTAKE-CYCLE+CORRECT and MISTAKE-CORRECT—improve the most. The worst-performing variants are NO-CYCLE and STUDENT-ONLY. The best variant, MISTAKE-CYCLE+CORRECT, improves by ~**9%** (**40.83%** → **44.43%**).

### 5.3 MISCONCEPTION INFERENCE

We see similar trends for the misconception inference task as we do for student simulation. As shown in Figure 3b, we find all MISTAKE variants lead to improvements in the MAP@k score, with MISTAKE-CYCLE+CORRECT leading to the best performance (**0.178** → **0.204**, representing a ~**15%** improvement over the pretrained model. Again, we find that only training the misconception model, *i.e.,* MISCONCEPTION-ONLY, leads to the worst performance.

| Model | Task Accuracy (%) | Student Simulation Accuracy (%) | Misconception Inference MAP@25 |
|---|---|---|---|
| MISTAKE + Llama-3.1-3B-Instruct | 69.4[†] | 44.4 | 0.204 |
| GPT-3.5-turbo | 74.1 | 40.6 | 0.206 |
| GPT-4o | 85.0 | 64.1 | 0.259 |
| GPT-4.1 | 92.4 | 66.3 | 0.271 |

Table 1: Comparison of the best models trained with MISTAKE with results from larger, closed-source GPT models. [†]Indicates that the result is reported from the pretrained Llama-3.1-8B-Instruct model. All other results are the best values achieved by a MISTAKE variant on the test set (see Figure 3 for full performance across rounds).

| Model | Base model | +MISTAKE-GENERATE | |
|---|---|---|---|
| Llama-3.1-8B-Instruct | 0.226 | **0.371** | (+0.145) |
| Qwen3-8B | 0.377 | **0.446** | (+0.069) |
| GPT-3.5-turbo | 0.320 | **0.375** | (+0.055) |
| GPT-4o | 0.427 | **0.497** | (+0.070) |
| GPT-4.1 | 0.447 | **0.490** | (+0.043) |

Table 2: Comparison of distractor precision for a variety of base models, with and without the cycle consistency filter condition in MISTAKE. See Figure 3 for results for other filtering conditions.

## 5.4 DISTRACTOR GENERATION

Figure 3c shows the precision of generated distractor answers for each question in the validation dataset for models trained on the MISTAKE-CYCLE+CORRECT data. We compare multiple sets of generated distractor answers. **UNCONDITIONAL** evaluates the answers generated by Sample_Answers in MISTAKE-GENERATE. We also evaluate the answers output by Simulate_Student in MISTAKE-GENERATE: **SIMULATED (NO FILTER)** evaluates all of the generated answers. **SIMULATED + CORRECT ANSWERS** only evaluates answers that are not equal to the correct answer, while **SIMULATED + CYCLE CONSISTENCY** is the full cycle consistency condition in MISTAKE-GENERATE, *i.e.,* only evaluating answers that are the same as original sampled answers.

We find that the simulated methods with filtering outperform UNCONDITIONAL and SIMULATED (NO FILTER) methods, suggesting that the procedure in MISTAKE-GENERATE of inferring misconceptions and simulating answers is effective at generating high-quality distractor answers. The distractors generated by SIMULATED + CYCLE CONSISTENCY are consistently the most aligned with the ground truth distractors than the other methods, suggesting that the cycle consistency check in particular is an effective way of improving the quality of generated distractors. The biggest improvement in distractor precision, with SIMULATED + CYCLE CONSISTENCY leading to a **64.6%** improvement over UNCONDITIONAL (**22.56%** → **37.14%**).

In addition, as shown in Table 2, applying the full SIMULATED + CYCLE CONSISTENCY filter in MISTAKE-GENERATE leads to improvements in distractor precision across all models we evaluate, including the most powerful models GPT-4o and GPT-4.1.

## 6 CONCLUSION

Overall, our experiments demonstrate that MISTAKE is an effective approach for modeling incorrect reasoning and that it leads to improved performance on three educational tasks, student simulation (§5.2), misconception inference (§5.3), and distractor generation (§5.4). We show that the cycle consistency check in MISTAKE-GENERATE and the joint training of student simulation and misconception inference models in MISTAKE-UPDATE are both key components of this procedure. Taken together, these results highlight that while modeling incorrect reasoning is challenging for existing models, MISTAKE is an effective first step towards this goal. Future work can explore how the models trained by MISTAKE can be used downstream in educational applications, *e.g.,* in conjunction with chat-based LLMs to provide tutoring tailored to misconceptions. Another interesting direction for future work is to explore how the cycle consistency conditions in MISTAKE can be used to create better user simulators in other settings such as chat-based tutoring or even non-educational domains where users' behaviors may be explained by misconceptions or latent cognitive patterns.

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
