756
757
758
759
760
761
762
763
764
765
766
767
768
769
770
771
772
773
774
775
776
777
778
779
780
781
782
783
784
785
786
787
788
789
790
791
792
793
794
795
796
797
798
799
800
801
802
803
804
805
806
807
808
809

---

You are an AI assistant designed to judge whether two answers to a math problem are conceptually the same or different. You will be shown a K-12 math problem and two answers to the problem. Please output 'Same' if the two answers are conceptually equivalent and just formatted differently. Please output 'Different' if the two answers are conceptually different. If one of the answers is the empty string and the other is not, you should output 'Different'. Please do not actually solve the problem. You should only judge whether the two answers are the same or different. Your output should be formatted as follows:

Judgment: [Same/Different]

Here are several examples to guide your answer:

Math Problem:'''Subject: Fractions of an Amount Question: How do you say this fraction? $\frac{3}{5}$

Answer 1: 3/5
Answer 2: Three fives
Judgment: Same

Math Problem:'''Subject: Multiplying and Dividing with Decimals
Question: $50.09 \div 0.1 =$'''

Answer 1: $500.09$
Answer 2: $500.9$
Judgment: Different

Math Problem:'''Subject: Properties of Polygons
Question: Tom and Katie are arguing about parallelograms. Tom says this shape is a parallelogram  Katie says this shape is a parallelogram  Who is correct?'''
Answer 1: Katie
Answer 2: Only Katie

Judgment: Same
Math Problem:'''Subject: Percentages of an Amount Question: What is $120\%$ of $50$ ?'''

Answer 1: $10$
Answer 2: $60$
Judgment: Different

Math Problem:'''Subject: Converting between Fractions and Percentages
Question: Convert this fraction to a percentage $\frac{4}{5}$'''

Answer 1: $45\%$
Answer 2: $45$
Judgment: Same

---

Table 2: Prompt to the GPT-4o-mini model for judging answer equivalence for the distractor generation evaluations.

| | |
|---|---|
| System | You are an AI assistant designed to infer incorrect answers that students might give in solving K-12 math problems so that they can be used as distracter choices for multiple choice problems. You will be shown a math problem. Please output 3 incorrect answers. Your output should start with 'Incorrect Student Answers:' followed by a list of 3 incorrect answers that a student might give in solving the problem and that could be used as distracter choices for the question. Each item should have the same formatting as the correct answer, but make sure none of the incorrect answers you output are the exact same as the correct answer. Make sure to output exactly 3 incorrect answers. Do not output any other information.

Here are several examples of math problems and misconceptions for student answers. Please use these examples to guide your future responses.

Math Problem:'''Subject: Adding and Subtracting Negative Numbers
Question: $6 - (-8) =$'''
Correct Answer: $14$
Incorrect Student Answers:
- $-14$
- $2$
- $-2$

Math Problem:'''Subject: Multiplying and Dividing Negative Numbers
Question: $-4 + (-5) \times 3 =$'''
Correct Answer: $27$
Incorrect Student Answers:
- $27$
- $-27$
- $-11$

...

Math Problem:'''Subject: Fractions of an Amount
Question: Here is a number card $\frac{1}{3}$ of $A$ is 12 What is $\frac{1}{2}$ of $A$ ?
Correct Answer: $2$

Incorrect Student Answers:
- $2$
- $8$
- $36$

Math Problem:'''Subject: Expanding Single Brackets
Question: The area of these shapes are equal. Which of these is correct? '''
Correct Answer: $12x - 8 = 10x + 4$

Incorrect Student Answers:
- $12x - 2 = 10x + 2$
- $8 + 6x - 4 = 4 + 10x + 4$
- $4 + 3x - 2 = 2 + 5x + 2$ |
| User | Math Problem:'''Subject: **Adding and Subtracting Fractions**
Question: **Work out:** $\frac{4}{11} + \frac{7}{11}$. **Write your answer in its simplest form.**'''
Correct Answer: **1** |
| Assistant | **Incorrect Student Answers:**
- $\frac{11}{22}$
- $\frac{11}{11}$
- $\frac{4}{11} - \frac{7}{11}$ |

Table 3: Example prompt/output for sampling incorrect answers with Sample_Answers. Bolded indicates variables that change.

864
865
866
867
868
869
870
871
872
873
874
875
876
877
878
879
880
881
882
883
884
885
886
887
888
889
890
891
892
893
894
895
896
897
898
899
900
901
902
903
904
905
906
907
908
909
910
911
912
913
914
915
916
917

You are an AI assistant designed to simulate student misconceptions in solving K-12 math problems. You will be shown a problem and a misconception a student has in solving the problem. You should simulate what answer a student with that misconception would give. Your output should be the answer that the student would give. Please do NOT generate the correct answer. Instead, solve the problem as if you had the misconception that the student has.

Please start your output by thinking step-by-step to reason through what answer a student with the misconception that you were shown would give. The step-by-step reasoning should start with "Reasoning:".

Then, on a new line, output 'Incorrect Student Answer:' and generate the answer that the student would give. When outputting 'Incorrect Student Answer', please only output the letter of the multiple choice answer that the student would give and nothing else. Even if none of the multiple choice answers seem to be the right choice, output the letter of the closest answer.

Here are several examples of math problems, misconceptions, and reasoning paths for student answers. Please use these examples to guide your future responses.

Math Problem:'''Subject: Range and Interquartile Range from a List of Data
Question: What is the range of the following numbers? [0.2,  0.22,  0.212]
Answer Choices:
A) 0.02
B) 0.012
C) 0.21
D) 0.2'''
Student Misconception: When calculating the range does not reorder the data to find the largest number minus the smallest number

Reasoning: Based on the student's misconception, we will assume they will not reorder the data and instead treat the first and last numbers in the list as the smallest and largest numbers. Then in order to calculate the range, they will subtract the smallest number from the largest number, i.e. 0.212-0.2=0.012. Thus, the student will choose answer choice B.
Incorrect Student Answer: B

Math Problem:'''Subject: Gradient as change in y over change in x
Question: What is the gradient of this line? 
Answer Choices: A) 0
B) 2
C) $x$
D) 1'''
Student Misconception: Believes the gradient of a line is given by an intercept

Reasoning: If the student thinks the gradient of the line is given by the intercept, they will think 0 is the gradient because the intercept is 0. Therefore, they will choose A.
Incorrect Student Answer: A

...

Table 4: Prompt used for the Simulate_Student module and student simulation evaluation experiments.

You are an AI assistant designed to infer misconceptions that students might have in
solving K-12 math problems. You will be shown a math problem, along with an incorrect
answer given by a student. Please output a 1-sentence description of the misconception
that a student who gave that incorrect answer might have. Make sure the misconception
is specific enough that it could be used to infer the incorrect answer given by the
student but general enough that it could apply to other similar problems.

Your output should be formatted as follows:
Reasoning: [reasoning for the misconception]
Misconception: [1-sentence description of the misconception]
Do not output any other information.

Here are several examples of math problems and misconceptions for student answers.
Please use these examples to guide your future responses.

Math Problem:'''Subject: Range and Interquartile Range from a List of Data
Question: What is the range of the following numbers? [0.2,   0.22,   0.212]

Answer Choices:
A) 0.02
B) 0.012
C) 0.21
D) 0.2'''
Incorrect Student Answer: B

Reasoning: The student's answer, B, or 0.012, corresponds to the difference 0.212
- 0.2. This is the difference between the first and last numbers. Therefore, the
student's misconception appears to be that they are not reordering the data to calculate
the range.

Misconception: When calculating the range does not reorder the data to find the largest
number minus the smallest number

Math Problem:'''Subject: Gradient as change in y over change in x
Question: What is the gradient of this line? ![A set of axes with a graph drawn on.
The graph passes through the following points: (-3,-3) (-2,-2) (-1,-1) (0,0) (1,1)
(2,2) and (3,3).]()
Answer Choices:
A) 0
B) 2
C) $x$
D) 1'''
Incorrect Student Answer: A

Reasoning: The student answers A, or 0, and the x and y intercepts of the given line
are also both 0. Therefore, the student's misconception may be about confusing the
gradient of the line with one of the intercepts.

Misconception: Believes the gradient of a line is given by an intercept

Math Problem:'''Subject: Length Units
Question: 450  cm= \square  m

Answer Choices:
A) 4.5
B) 45
C) 5
D) 45000'''
Incorrect Student Answer: C

Reasoning: To find the value of the square, the correct approach would be to convert
$[450cm]$ to meters. There are 100 centimeters in a meter, and so the correct calculation
is $[450 \div 100 = 4.5m]$. However, the student answers C, or 5, which suggests that they
may be rounding up to the nearest whole number to give their answer.

Misconception: Gives a rounded whole number instead of a decimal
...

Table 5: Prompt used for the `Infer_Misconception` module and for the misconception inference
experiments.

| Variant | Values of $\alpha$ when examples are: | |
| --- | --- | --- |
| | cycle consistent $(s = a)$ | correct answer $(s = a^*)$ |
| MISTAKE-CYCLE+CORRECT | **2** | **0** |
| MISTAKE-CORRECT | 1 | **0** |
| NO-CYCLE | 1 | 1 |

Table 6: Variants of MISTAKE and the values of $\alpha$ that they use, where $\alpha$ the weight given to each example. $\alpha = 1$ is the default value.

---

```
Model outputs for student simulation (validation dataset, round 4).

Math Problem:'''Subject: Volume and Capacity Units
Question: 3 litres = ⋆ millilitres
Answer Choices: A) 0.3   B) 30   C) 300   D) 3000'''

Student Misconception: Thinks there are 100 ml in a litre

Model Output: Reasoning: Assumes 100 ml per litre, so 3 × 100 = 300. Chooses C.

Incorrect Student Answer: C

Actual Output: C
```

---

```
Math Problem:'''Subject: Multiples and Lowest Common Multiple
Question: Which of the following numbers is a common multiple of 4, 6, and 12?
Answer Choices: A) 2   B) 12   C) 18   D) 20'''

Student Misconception: Identifies a multiple of a number but not a common multiple of
a group

Model Output: Reasoning: Thinks 12 is a multiple of 4, 6, and 12. Chooses B.

Incorrect Student Answer: B

Actual Output: D
```

---

Table 7: Example model outputs for student simulation from the validation dataset in round 4.

```
Model outputs for misconception inference (validation dataset, round 4).

Math Problem:'''Subject: Expanding Double Brackets
Question: What should replace the circle when these two brackets are expanded and
simplified?
```
$(p - 3)(p + 1) \equiv p^2 \triangle p \bigcirc$
```
Answer Choices: A) ◯ = −3    B) ◯ = +3    C) ◯ = −4    D) ◯ = −2'''

Incorrect Student Answer: B (+3)

Model Inferred Misconception: Neglects to distribute the negative sign when expanding
brackets

Actual Misconception: Believes multiplying a positive by a negative gives a positive
answer
```

```
Math Problem:'''Subject: Adding and Subtracting with Decimals
Question: 50.09 − 0.1 =
Answer Choices: A) 49.99    B) 50.99    C) 50.08    D) 38.98'''

Incorrect Student Answer: B (50.99)

Model Inferred Misconception: Thinks subtraction is performed as addition

Actual Misconception: When borrowing in subtraction, does not subtract one from the
preceding digit
```

Table 8: Example model outputs for misconception inference from the validation dataset in round 4.