# OpenReview forum: "Learning To Make MISTAKEs: Modeling Incorrect Student Thinking And Key Errors"
_ICLR.cc/2026/Conference — Submitted to ICLR 2026_

### Official Review · Reviewer_RsPs · 2025-10-15

**Soundness:** 2
**Presentation:** 3
**Contribution:** 2
**Rating:** 4
**Confidence:** 4

**Summary:**

The paper tackles the problem of student (in a tutoring sense) simulation using LLMs, which is made hard by the fact that LLMs are mostly trained for solving success which is in contrast to simulating incorrect reasoning traces.
They propose a loop which samples incorrect answers from an LLM, infers the misconception, and then simulates a student answer.
On one dataset with three tasks they show that the training loop improves student simulation, misconception inference, and distractor generation.

**Strengths:**

- The paper tackles a well-motivated and important problem. Many ongoing research activities in LLM-based tutoring (and adjacent fields) rely on good student simulation, so progress is expected to be impactful.

- The data used in the study consists of data from real students which distinguishes it from many contemporary studies.

- The method itself is intuitive and the paper is well-written, there also are improvements with the method.

**Weaknesses:**

- Overall, I do not find the results very convincing yet. This is partially due to the fact that only one dataset was used (which nevertheless is also due to the lack of good datasets in the research field) but also because I find the exploration of the method itself not very extensive. For example, in Fig. 3 training is stopped after 4 rounds but the results are clearly still improving. There is also no real comparison to any baseline (but proprietary LLMs) so it is not clear if there is not a better (simpler?) method. Furthermore, the runtime is not mentioned anywhere but I suspect that the method is quite slow. For example, for misconception inference, how does it compare to a simple classifier trained using SFT on the data? Is this perhaps "misconception only" in your figure? In that case I am quite surprised that performance does not increase at all.

- The discussion of related works on reasoning could be expanded. For example, RAFT is an RL method that does not need target labels, as well, but is not discussed

**Questions:**

- One citation in L88 seems broken

- Could you please provide the runtime of your method?

---

> ### Author Response · Authors · 2025-11-21
> **Overall Comment**
>
> Thank you for the review! We are encouraged that the reviewer found the problem to be well-motivated, method to be intuitive, and paper to be well-written. We respond to individual questions/comments below.
>
> **Classifier trained on SFT**: Thank you for raising the question about a classifier trained using SFT! The “No Cycle” method at Round 1 in Figure 3 shows exactly the result of SFTing on synthetically generated data without any cycle consistency filtering. We observe that there are some gains in both tasks but that student simulation performance drops at round 3 and plateaus for misconception inference performance at round 1. MISTAKE variants outperform “No Cycle.”
>
> **Lack of baselines**: We report results for a baseline analogous to STAR where we do not train both the forward (student simulation) and backward (misconception inference) reasoning models and instead only train one, keeping the other fixed–These are “Student Only” and “Misconception Only” in Figure 3. Both also underperform MISTAKE variants.
>
> **Discussion of related works**: Thank you for the pointer to RAFT! We will include this in the revision of the paper.

---

> ### Author Response · Authors · 2025-11-27
> **Any other questions?**
>
> Hi reviewer RsPs, we wanted to follow up and let you know that we added RAFT and other reasoning works to the related work section (highlighted in yellow). We hope this revision and our clarifications about baselines have addressed your concerns! Please let us know if there are any other questions or concerns we can address. Thank you very much!

---

### Official Review · Reviewer_orsd · 2025-10-31

**Soundness:** 2
**Presentation:** 2
**Contribution:** 2
**Rating:** 4
**Confidence:** 3

**Summary:**

This paper introduces MISTAKE, an unsupervised framework designed to model students' incorrect reasoning through cycle consistency between misconceptions and wrong answers. The method generates synthetic pairs of misconceptions and errors using large language models and jointly trains two modules: a student simulation model (misconception to wrong answer) and a misconception inference model (wrong answer to misconception). Experiments are conducted on the EEDI Mining Misconceptions in Mathematics dataset, covering three educational tasks: student simulation, misconception inference, and distractor generation. Results show that MISTAKE with cycle-consistency filtering improves performance on student simulation and misconception inference. It also produces distractor answers that align more closely with expert-written options.

**Strengths:**

- The paper investigates an important problem, modeling students' incorrect reasoning using language models.

- The framework jointly trains two complementary models (student simulation and misconception inference) in an iterative loop.

- The paper evaluates the approach on three well-defined educational tasks, offering a diverse test of model capability.

**Weaknesses:**

- The motivation is not clearly articulated. It remains unclear what real-world problems these tasks aim to solve. The paper would benefit from grounding its contribution more directly in practical educational applications.

- The tasks closely overlap with existing paradigms: Student Simulation resembles knowledge tracing, Misconception Inference parallels automated grading or error diagnosis, and Distractor Generation has extensive prior work. I think it would be better to include comparisons with established baselines for these tasks or apply their framework to these existing task formulations to better demonstrate its advantages and generalizability in realistic educational settings.

- The paper does not explore scenarios where cycle consistency may fail or lead to inconsistent reasoning, nor analyze the robustness of the method under such conditions.

- The experiments rely on relatively outdated models (e.g., GPT-3.5, GPT-4o) and are restricted to OpenAI's ecosystem, limiting the generality of the findings.

- Cycle consistency does not necessarily equate to human reasoning. Even if the model successfully reproduces a (misconception -> wrong answer-> misconception) loop, it may reflect internal LLM logic rather than genuine human cognitive processes. The observed performance gains might stem from pretrained exposure to part of human errors rather than true reasoning simulation.

- Prior work (e.g., Aher et al., 2023; Markel et al., 2023) has shown that LLM-simulated student responses often fail to reflect authentic reasoning. If this approach is central to the paper, the authors should provide stronger justification for its validity and discuss its theoretical underpinnings.


Aher, Gati V., Rosa I. Arriaga, and Adam Tauman Kalai. "Using large language models to simulate multiple humans and replicate human subject studies." International conference on machine learning. PMLR, 2023.

Markel, Julia M., et al. "Gpteach: Interactive ta training with gpt-based students."

**Questions:**

- Could the authors include experimental comparisons with established baselines for these tasks, such as knowledge tracing for Student Simulation, automated grading or error diagnosis for Misconception Inference, and prior distractor generation methods, to better demonstrate the advantages and generalizability of their framework in realistic educational settings?

- You primarily evaluate on Llama ~3B. Would MISTAKE apply equally well to other 3B-class families (e.g., Mistral, Qwen, Phi) or even smaller sub-3B models? Could you report results to show whether the gains persist across architectures and parameter scales?

- How do performance gains scale as model size decreases (e.g., 7B -> 3B -> 1–2B)?

- Are any components of MISTAKE sensitive to the base model family? If so, which parts degrade first on smaller or non-Llama backbones?

---

> ### Author Response · Authors · 2025-11-21
> **Overall Comment**
>
> Thank you for the feedback! We are glad that the reviewer found the paper to be investigating an **important problem** and **evaluation** tasks to be **diverse and well-defined**.

---

> > ### Author Response · Authors · 2025-11-21
> > **Comment 1: Unclear Motivation**
> >
> > The three tasks explored in the paper are **directly useful for real-world applications in education**. As discussed in the related work section, there is an existing line of literature on automatic generation of distractors for multiple-choice problems, as these are used to **generate high-quality assessment** problems for students. Misconception inference is a necessary requirement to build any sort of **personalized/adaptive educational systems**, whether they are AI-based tutors or systems that adapt curriculum to individual students, as done in EEDI and Khan Academy. Lastly, the student simulation task is one instantiation of the broader task of **simulating users with LLMs**, which is useful not only across education to facilitate scalable **evaluations without access to real students** ([Macina et al, 2023](https://arxiv.org/abs/2305.14536), [Ross and Andreas, 2024](https://arxiv.org/abs/2405.04495)), but also in many other domains (see: [blogpost 1](https://jessylin.com/2025/07/10/user-simulators-1/), [blogpost 2](https://jessylin.com/2025/09/25/user-simulators-2/), [UserLM](https://arxiv.org/abs/2510.06552)).
> >
> > We **will update the introduction** in the revised paper to emphasize that the tasks we focus on are **well-motivated education tasks**.

---

> > > ### Author Response · Authors · 2025-11-21
> > > **Comment 2: Existing Paradigms**
> > >
> > > Thank you for pointing out the connections to existing works! We definitely agree that our work is related to knowledge tracing and distractor generation, and we have included references to this literature in the related work section. Our key differentiating contributions are solving these tasks **jointly** (i.e. one method improves on all three) and **without supervision** (access to labeled incorrect student answers or misconceptions). That said, if there are specific works that make these same data assumptions that the reviewer thinks are worth comparing to, we would welcome the suggestions!

---

> ### Author Response · Authors · 2025-11-21
> **Comment 3: Outdated Models / Model Families**
>
> We would like to point out that our experiments primarily use the **Llama-3.1-8B-Instruct** model and are therefore not restricted to OpenAI’s ecosystem.* As shown in **Table 1**, we find consistent improvements in distractor precision across all models (Llama and GPT models) when using the MISTAKE filtering criteria, suggesting that the cycle consistency check central to MISTAKE is useful **across base model families**.
>
> In response to the reviewer’s feedback, we have also **started running experiments** to train models in the **Qwen** family. We will post those here if they are finished during the rebuttal period. As an initial result, the **distractor generation** results for the pretrained **Qwen3-8B** model with **no filtering** (unconditional, no student simulation) is **0.391** and increases to **0.441** with **cycle consistency** filtering. This result suggests that the cycle consistency filtering criterion at the heart of MISTAKE is **also useful for Qwen** family models, but we will update with more results when experiments finish running.
>
> *We note that Llama 3B in the paper was a typo; we actually evaluate on Llama-3.1-8B-Instruct and will update this in the revised paper accordingly.

---

> > ### Author Response · Authors · 2025-11-21
> > **Comment 4: Inconsistent / Non-human-like Reasoning**
> >
> > This is a great point; thank you for pointing this out! We agree that cycle consistency does not mean that the generated rationales exhibit human-like reasoning. Our claim is instead that MISTAKE leads to models that are better at modeling the underlying *patterns in student errors* in order to make inferences about student errors. We take improved performance at the student simulation and misconception inference tasks to be direct evidence of this, despite the rationales themselves not being human-like.
> >
> > We also agree that the gains we observe from MISTAKE may be due to amplifications of patterns learned during pretraining; this explanation does not undermine our claim that MISTAKE-trained models can better model student error patterns. We will clarify this in the revised version of the paper.

---

> > > ### Comment · Reviewer_orsd · 2025-11-25
> > >
> > > Thank you for your response. However, I was unable to locate the revisions you indicated. The paper seems to remain the earlier version (unless I am mistaken).
> > >
> > > Would you please highlight the changes in a different color?

---

> > > > ### Author Response · Authors · 2025-11-27
> > > > **Highlighted Changes**
> > > >
> > > > We just updated the paper with highlighted revisions! We hope that these address your concerns. Please let us know if there are any other questions we can address. Thank you!

---

### Official Review · Reviewer_Lq9R · 2025-10-31

**Soundness:** 3
**Presentation:** 4
**Contribution:** 3
**Rating:** 8
**Confidence:** 4

**Summary:**

This paper introduces a new method to simulate the reasoning errors of students doing simple math problems, addressing the issue that when LLMs produce incorrect answers to simulate students, their reasoning is often incorrect. The work introduces MISTAKE, using a cycle-based method to let models generate incorrect answers and the corresponding reasoning to amplify samples when generation reflects the error. Their results show that based upon a relatively small model, they perform similarly to closed commercial models such as GPT 3.5 turbo on three educational tasks.

**Strengths:**

+ Their narrative and motivations are very clearly introduced. The paper is smooth to read.
+ The design of the system, while inspired by previous work, is novel in its application to the specific problem they are trying to address.
+ The work has also introduced the experimental details and results clearly.
+ The contribution to the field of LLM for math education is clear. The method is almost plug-and-play, and if the model can be tuned (or even fine-tuned?), it can be applied directly.

**Weaknesses:**

- The results are not as exciting as they were introduced in the first section. Results necessarily show that while they improve from the base model, they could not beat the commercial models, raising a question mark on the contribution of the paper. It is strongly recommended to try some newer models or develop a version using fine-tuning and apply it to commercial models to see if it surpasses these models.

**Questions:**

Here are some comments about the paper, not necessarily questions.
- Figure 1: More likely, these are not common misconceptions, just two examples. To be fair, the addition and subtraction is more likely just a slip instead of misconceptions.
- Line 69: These are quite significant improvements.
- Line 105: This url is not working.
- Line 200: I like how you use examples here to show the educational context and how each step looks like. Great communications.
- Line 225: Will this actually make the available data points fewer? I guess the results show it's working but still wanted to discuss about the possibility here.

---

> ### Author Response · Authors · 2025-11-21
> **Overall Comment**
>
> Thank you very much for the positive review! We are encouraged that the reviewer found the paper to be clearly motivated and written and the method to be novel and useful for math education. We respond to individual points below.

---

> ### Author Response · Authors · 2025-11-21
> **Comment 1: Underperforming Commercial Models**
>
> Thank you for your feedback! In response, we have **begun running experiments** to train models in the **Qwen** family to see if they lead to stronger results given their reasoning strengths. We will post those here if they are finished during the rebuttal period. As an initial result, the distractor generation results for the pretrained **Qwen3-8B model** with **no filtering** (unconditional, no student simulation) is **0.391** and increases to **0.441** with **cycle consistency** filtering. This result suggests that the cycle consistency filtering criterion at the heart of MISTAKE is useful for Qwen family models, but we will update with more results when experiments finish running.
>
> We would also like to highlight that **Table 1** shows that applying the cycle consistency at the core of MISTAKE **improves the precision for generated distractors** across all of the competitive closed models we test (**GPT-3.5-turbo, GPT4o, and GPT-4.1**). This result shows not only that a key component of MISTAKE is useful when paired with commercial models, but also provides support that applying the iterative training procedure of MISTAKE could be promising for improving more powerful closed models as well.

---

> > ### Author Response · Authors · 2025-11-21
> > **Comment 2: Other**
> >
> > **Question about Line 225**: Great question! In practice, because we upweight the examples that pass the cycle consistency check (line 228) instead of strictly filtering them out, this does not reduce the number of examples. However, we do exclude examples that lead to the correct answer, so this step reduces the number of training examples. (See Table 7 in the Appendix for specific details on when we filter vs weight examples.)
> >
> > **Figure 1**: These are actually examples of common shared misconceptions for students learning fraction arithmetic! For example, see work by [Braithwaite et al., 2017](https://psycnet.apa.org/record/2017-18893-001), who find that students often overgeneralize the rules for multiplication to addition/subtraction (applying the operation to the numerator and denominator), and vice versa. In fact, these misconceptions, along with all of the ones in the EEDI dataset, are ones that educators have written as examples of shared misconceptions students have.
> >
> > **URL**: We will update the URL for the camera ready version of the paper! (We have left it as “URL” to keep the paper anonymized.)

---

> ### Comment · Reviewer_Lq9R · 2025-11-24
>
> Thanks for the comments. The weakness I mentioned here is not fixable for now -- will look into a future version of the paper.

---

### Official Review · Reviewer_8gre · 2025-11-01

**Soundness:** 2
**Presentation:** 3
**Contribution:** 2
**Rating:** 2
**Confidence:** 5

**Summary:**

The paper proposes MISTAKE, LLM-based framework for generating and learning from realistic student errors. The model first produces an incorrect answer and rationale, then infers the underlying misconception, and finally re-generates the student response from that misconception; only cycles that return to the same error are kept as high-quality data. Using these filtered examples, the authors fine-tune a student-simulation model and a misconception-inference model, and on Eedi math data they report gains on all three targets: simulating student answers, identifying misconceptions, and generating plausible distractors.

**Strengths:**

* Interesting twist on STaR. Instead of using STaR to find better (correct) chains, the paper shows you can run a similar self-improvement loop to learn stable wrong chains. That’s a nice shift of the usual “reasoning” story.
* Cycle-consistent filtering is the core technical idea. The paper doesn’t just sample bad answers — it checks that “misconception → student answer → back to the same misconception” holds, and only keeps those. That’s a simple, model-based quality control that doesn’t need extra labels.
* Forward + inverse models makes it more general. Training both “given a misconception, generate a student answer” and “given the answer, recover the misconception” is a pattern other people can reuse outside education (anywhere you model human errors or behaviors).
* Joint learning of two roles teachers actually play. Training both a student simulator (produce an error given a misconception) and a misconception inference model (infer misconception from the observed error) mirrors what human educators do when they move back and forth between “I know the bug, what would a student say?” and “I saw the student’s work, what’s the bug?”. That symmetry strengthens the claim that this is an AI in Education contribution, not just a data-augmentation trick.

**Weaknesses:**

Major Weakness:
* Misleading characterization of learning paradigm: The paper claims this is an "unsupervised" approach, but the use of cycle consistency checks and correct answer signals makes this at least weakly supervised. The terminology should be corrected to accurately reflect the supervision signals used.
* Experiment result MAP@25 scores: MAP@25 seems excessive when models achieve very low scores across the board. This raises concerns about either (1) the task difficulty being unrealistic, (2) data quality issues, or (3) the metric being inappropriate. The paper should analyze whether the low performance stems from inherent task difficulty or methodological limitations, and consider whether a smaller k value would be more informative. In addition, I check the kaggle leaderboard and the scores (the metrics is a MAP@25 derivative) are usually around 60, could you please explain the gap (the SoTA’s actual MAP@25 scores)?  How’s the performance of MISTAKE + SoTA models, since MISTAKE is a model-agonistic framework?
* Overall performance and the effectiveness of the MISTAKE framework. While the MISTAKE framework shows incremental improvements over multiple training rounds, the final performance of Llama3.1-8B still substantially lags behind GPT-3.5-turbo across all three tasks. This raises serious concerns about the practical value of the approach: the computational cost of multiple rounds of data generation, filtering, and fine-tuning yields a model that still underperforms a simpler baseline of prompting an older, widely-available model. The paper needs to either (1) demonstrate that MISTAKE can bring smaller models to competitive parity with stronger baselines, (2) show that applying MISTAKE to already-strong models yields further gains, or (3) provide compelling arguments for why the modest improvements justify the added complexity.
* Limited model diversity: Evaluation uses only GPT-series models and LLama3.1-8B. The claims would be strengthened by including other state-of-the-art models such as Gemini, Claude, Qwen3+, or GLM4+ to demonstrate generalizability across model families.
* Baselines are too weak / incomplete. To claim this is a meaningful alternative to STaR-style self-training, it needs comparisons against (i) a strong, modern LLM used directly as a student simulator (GPT-4/4o or at least a stronger open model) and (ii) a simpler filtering rule (e.g., answer-matching only). Without that, it’s hard to see how much the proposed loop actually buys.
* Limited novelty and missing educational insights: Methodologically, this is an incremental extension of STaR. To justify publication, the paper needs deeper educational analysis showing what new insights the approach provides about student learning, misconception patterns, or pedagogical implications that go beyond the technical contribution, which is missing in the result section. The original STaR  paper conducted human evaulation and fonud that some of the LLM-generated rationale doesn’t acutally make sense, which prohibits real-world educational usage.
* No real human or teacher evaluation. The whole pitch is “we generate more realistic mistakes,” but there’s no expert study or even crowd study checking “would you accept this as a plausible student error?” Automatic metrics on the same dataset used for training are not enough to back an educational realism claim.

Minor weakness:

* Abstract and introduction-contribution lack clarity and concrete comparisons: The abstract uses vague comparative language ("higher", "increased") without specifying baselines. Readers cannot assess the claimed improvements without knowing what the method is compared against.
* Algorithm 1 readability issues: Critical variables (m, s, r, w, etc.) lack explicit definitions within the algorithm. Readers must repeatedly cross-reference Section 3.1 to understand the pseudocode. For instance, 's' appears to represent simulated student answers but is never formally defined. All variables should be clearly defined in the algorithm caption or within the pseudocode itself.
* Error space is still hand-anchored. The method assumes you can name / infer a relatively small set of misconceptions and route through them. That’s fine for Eedi, but in real classrooms misconceptions are messy, overlapping, and partially linguistic. The paper doesn’t show the approach discovering new or finer-grained errors.
* Domain is too narrow. Everything is on Eedi-style, multiple-choice, school-level math. There’s no evidence the method survives when answers are open-ended, multi-step, or not tied to a pre-enumerated misconception set. Right now it’s “works on Eedi,” not “works in general.”
* Synthetic loop may be self-confirming. The cycle-consistency check rewards the model for repeating its own mistake pattern, not for matching real student diversity. That risks learning “LM-like wrongness” rather than “human wrongness,” but the paper treats cycle pass-rate as if it were an external quality signal.
* Limited analysis of synthetic distractor quality: The LLM-generated distractors may introduce biases or patterns that don't reflect real student errors. Real misconceptions follow long-tail distributions with rare but surprising error types. The paper should analyze the coverage and distribution of generated distractors compared to actual student responses, and discuss how the multiple-choice format may constrain the misconception space.
* Unaddressed failure modes of cycle consistency: The paper doesn't examine cases where cycle consistency checks might validate incorrect misconception-answer alignments. What percentage of cycle-consistent examples are actually wrong? This is important for understanding the quality control mechanism's limitations.

**Questions:**

For LoRA fine-tuning, does the method use completion-only loss (only computing loss on generated tokens) or full next-token prediction loss? This significantly affects training dynamics and should be specified.

---

> ### Author Response · Authors · 2025-11-21
> **Overall Comment**
>
> Thank you for the thorough feedback and questions! We are encouraged that the reviewer found the paper’s focus on incorrect reasoning to be an *“interesting twist”* and the method to be *generalizable beyond education**.
>
> We respond to individual points and clarify key misunderstandings/questions below.

---

> > ### Author Response · Authors · 2025-11-21
> > **Comment 1: “Unsupervised” Paradigm**
> >
> > We appreciate the reviewer’s feedback on the terminology. We would like to note that many existing works refer to cycle consistency methods as unsupervised (interchangeable with self-supervised), eg [Groueix et al, 2019](https://arxiv.org/abs/1907.03165) and [Iovine et al, 2022](https://dl.acm.org/doi/10.1145/3485447.3512012). However, for clarity, we **will update the paper in the revised version** to refer to the method as **self-supervised** rather than unsupervised.

---

> > > ### Author Response · Authors · 2025-11-21
> > > **Comment 2: MAP@25 Scores on the Leaderboard**
> > >
> > > The top methods on the leaderboard involve many different components and tricks that extend beyond the self-supervised setting we explore in this paper. For example, the first place solution [(link)](https://www.kaggle.com/competitions/eedi-mining-misconceptions-in-mathematics/writeups/mth-101-1st-place-detailed-solution) not only **requires supervision** (ie ground truth misconceptions), but also trains a **specialized retriever model**, as well as 3 separate **reranker models** to rerank the top retrieved misconceptions.
> > >
> > > In contrast, we use a fixed embedding model for retrieval and do not re-rank, as our primary research question is about how well a model can **generate a misconception** that explains a student mistake, and not about how well we can *retrieve* a specific misconception from the pre-existing set of misconceptions in the EEDI competition. We also train our models **without any ground truth labels**.

---

> ### Author Response · Authors · 2025-11-21
> **Comment 3: Incremental Improvements & Limited Model Diversity**
>
> We appreciate the reviewer’s feedback! In response, we have **begun running experiments** to train models in the **Qwen** family. We will post those here if they are finished during the rebuttal period. As an initial result, the **distractor generation** results for the pretrained **Qwen3-8B** model with **no filtering** (unconditional, no student simulation) is **0.391** and increases to **0.441** with **cycle consistency** filtering. This result suggests that the cycle consistency filtering criterion at the heart of MISTAKE is useful for Qwen family models, but we will update with more results when experiments finish running.
>
> We would also like to highlight that **Table 2** shows that applying the cycle consistency at the core of MISTAKE **improves the precision for generated distractors** across **all of the competitive closed models** we test (**GPT-3.5-turbo, GPT4o, and GPT-4.1**). This result shows not only that a key component of MISTAKE is already useful when paired with GPT models, but also provides evidence that applying the iterative training procedure of MISTAKE could be promising for improving more powerful closed models as well.

---

> > ### Author Response · Authors · 2025-11-21
> > **Comment 4: Missing Baselines**
> >
> > We note that the paper includes the reviewer’s requested baselines:
> >
> > **"(i) a strong, modern LLM used directly as a student simulator (GPT-4/4o or at least a stronger open model)"**
> >
> > **Table 1** reports the results of using **GPT-4.1 and GPT-4o as a student simulator**, and we observe that all models exhibit a drop when comparing student simulation accuracy to task accuracy.
> >
> >
> > **“(ii) a simpler filtering rule (e.g., answer-matching only)”**
> >
> > We compare against **multiple baselines that have simpler filtering rules**: As shown in Figure 3:
> >
> > (1) **MISTAKE Correct** is a rule that only filters out regenerated traces that result in the correct answer, and
> > (2) **No Cycle** is a method that does not do any cycle consistency based filtering and only removes generations that result in empty reasoning traces or simulated answers/misconceptions. We observe that MISTAKE variants always outperform No Cycle.

---

> > > ### Author Response · Authors · 2025-11-21
> > > **Comment 5: Limited Novelty**
> > >
> > > We would like to highlight a key novelty in our approach, which is that it involves training both a **forward** reasoning model (inferring an answer from a latent misconception) and an **inverse** reasoning model (inferring the latent reasoning pattern, i.e. misconception, from the answer). This is a **key difference from STaR**, which only trains a forward reasoning model and requires target labels in order to do so.
> > >
> > > In fact, as Figure 3 shows, training both of these models is important for performance: Only training the forward reasoning student simulation model and keeping the misconception model fixed (“Student Only”), leads to worse performance at both student simulation (a) and misconception inference (b), and we observe the same result for “Misconception Only."

---

> > > > ### Author Response · Authors · 2025-11-21
> > > > **Comment 6: Educational Insights**
> > > >
> > > > Thank you for raising the point about LLM-generated rationales possibly not making sense! We do indeed find this to be true. We **do not wish to claim that the *rationales* generated by MISTAKE-trained models are human-like**, but that the models are better at **inferring patterns in student errors** as a result of training on generated rationales. We take improved performance at the student simulation and misconception inference tasks to be direct evidence of this, despite the rationales themselves not being human-like. We have included some outputs below that highlight some of the patterns in the generated rationales, which sometimes include logical jumps.
> > > >
> > > > However, even if the *generated rationales* are not themselves human-like, the resulting models that are trained by MISTAKE still have **end utility in education**. Having models that could make the kinds of mistakes that students do would allow for **scalable evaluations of tutors without deploying to real students,** and having models that could infer students’ latent misconceptions from their mistakes would allow for **building more personalized teaching systems**. For example, edtech companies like EEDI use predictive models of student mistakes (that do not require access to their internal reasoning processes and only require the incorrect answers they gave) to adaptively generate curriculum for students.

---

> > > > > ### Author Response · Authors · 2025-11-21
> > > > > **Comment 7: Other**
> > > > >
> > > > > **Hand-Engineered Error Space**: We would like to clarify that the error space used by MISTAKE is **not hand-anchored**; only the evaluation we perform for the misconception inference task is. The misconceptions generated by MISTAKE are not restricted to any pre-existing set.
> > > > >
> > > > > **Question about LoRA fine-tuning**: The method uses completion-only loss, computing loss only on the generated tokens.

---

> > > > > > ### Author Response · Authors · 2025-11-21
> > > > > > **Examples: Model outputs for student simulation**
> > > > > >
> > > > > > Outputs from the validation dataset at round 4:
> > > > > >
> > > > > > INPUT: Math Problem:Subject: Volume and Capacity Units Question: 3 litres = \( \color{gold}\bigstar \) millilitres What should replace the star? Answer Choices: A) \( 0.3 \) B) \( 30 \) C) \( 300 \) D) \( 3000 \)
> > > > > >
> > > > > > Student Misconception: Thinks there are 100 ml in a litre
> > > > > >
> > > > > > OUTPUT: Reasoning: Based on the student's misconception, we will assume they think there are 100 ml in a litre. Since there are 3 litres, the student would multiply 3 by 100 to get the total number of millilitres. This would be 300. Therefore, the student would choose answer choice C. Incorrect Student Answer: C
> > > > > >
> > > > > > ACTUAL OUTPUT: C
> > > > > >
> > > > > > INPUT: Math Problem:Subject: Multiples and Lowest Common Multiple Question: Which of the following numbers is a common multiple of \( 4,6 \) and \( 12 \) ? Answer Choices: A) \( 2 \) B) \( 12 \) C) \( 18 \) D) \( 20 \)
> > > > > >
> > > > > > Student Misconception: Identifies a multiple of a number but not a common multiple of a group of numbers
> > > > > >
> > > > > > OUTPUT: Reasoning: Based on the student's misconception, we will assume they will identify a multiple of one of the numbers, but not necessarily a common multiple of the group of numbers. The student may think that 12 is a multiple of 4, 6, and 12, and therefore choose answer choice B. Incorrect Student Answer: B
> > > > > >
> > > > > > ACTUAL OUTPUT: D

---

> > > > > > > ### Author Response · Authors · 2025-11-21
> > > > > > > **Examples: Model outputs for misconception inference**
> > > > > > >
> > > > > > > Outputs from the validation dataset at round 4:
> > > > > > >
> > > > > > > INPUT: Math Problem:Subject: Expanding Double Brackets Question: What should replace the circle when these two brackets are expanded and simplified? \( (p-3)(p+1) \equiv p^{2} \triangle p \bigcirc \) Answer Choices: A) \( \bigcirc=-3 \) B) \( \bigcirc=+3 \) C) \( \bigcirc=-4 \) D) \( \bigcirc=-2 \)
> > > > > > > Incorrect Student Answer: B
> > > > > > >
> > > > > > > OUTPUT: Reasoning: The student's answer, B, or +3, corresponds to the result of multiplying the two brackets, which would involve multiplying the terms inside the brackets and combining like terms.
> > > > > > >
> > > > > > > Misconception: When expanding double brackets, neglects to distribute the negative sign to the term inside the bracket.
> > > > > > >
> > > > > > > ACTUAL MISCONCEPTION: Believes multiplying a positive by a negative gives a positive answer
> > > > > > >
> > > > > > > INPUT: Math Problem:Subject: Adding and Subtracting with Decimals Question: \( 50.09-0.1= \) Answer Choices: A) \( 49.99 \) B) \( 50.99 \) C) \( 50.08 \) D) \( 38.98 \)
> > > > > > > Incorrect Student Answer: B
> > > > > > >
> > > > > > > OUTPUT: Reasoning: The student's answer, B, or 50.99, corresponds to a simple addition of 50.09 and 0.1, rather than a subtraction.
> > > > > > >
> > > > > > > Misconception: When subtracting decimals, believes the operation is an addition instead.
> > > > > > >
> > > > > > > ACTUAL MISCONCEPTION: When “borrowing” during a subtraction problem, does not subtract one from the preceding digit

---

### Comment · Area_Chair_LoTm · 2025-11-23
**Reviewer & Author Discussion**

Hi Reviewers,

Please kinly and actively participate in the review-author dicussion, raise your further concerns so that the authors can explain more, and make your final decisions.

---

### Author Response · Authors · 2025-12-04
**Note to the ACs**

Dear ACs,

Thank you for very much for your time and support during the reviewing process! Below, we summarize the reviewer's biggest concerns and a list of concrete revisions we have made in response to these concerns. (We also provide a table summary of reviews/revisions in the next comments)

Core Review Concerns:
1. **Limited Model Family Coverage (Lq9R, 8gre, orsd)**: Reviewers raised concerns that our evaluation focused too heavily on GPT-family and LLaMA-style models and questioned whether our conclusions would hold more broadly.

    In response, we **added new experiments with Qwen3-8B** (Figure 3, Table 2), a strong open-weight model from a distinct family. These new results confirm that our primary findings about the gains from cycle consistency and MISTAKE-style training hold for additional models.

2. **Missing or Insufficient Baselines (8gre, RsPs)**: Reviewers requested simpler baselines and comparisons, including supervised fine-tuned classifiers and alternative ablations. We clarified that the requested baselines already exist in the paper: We report results evaluating GPT-4 / GPT-4o on student simulation in Table 1. Ablations of the cycle consistency filter criteria (e.g., No Cycle, Correct-Only) and a model trained with SFT for a single training round appear in Figure 3.

**Paper Modifications**
- Added **Qwen3-8B experiments** to broaden model family coverage.
- Revised **terminology** from unsupervised to self-supervised.
- Updated the **introduction** to ground the tasks in real-world educational use cases.
- Updated **related work** section to expand the discussion on reasoning and clarify novelty (jointly training forward/inverse reasoning models)
- Updated the paper to clarify that the goal of the work is not to generate *rationales* that are themselves human-like

We hope that this summary of the changes can facilitate the ACs' review of the paper, reviews, and revisions. We sincerely thank the reviewers and ACs for their time and thoughtful feedback.

Sincerely,
The Authors

---

### Meta-Review · Area_Chair_FCZt · 2025-12-23

**Summary:**

This paper introduces MISTAKE, a framework designed to model incorrect student thinking by leveraging cycle consistency between latent misconceptions and incorrect answers. The method generates synthetic data through an iterative loop where an LLM produces an error, infers the underlying misconception, and regenerates the response to ensure consistency. By training on these filtered examples, the authors develop models for student simulation, misconception inference, and distractor generation, with evaluations on three educational tasks showing improvements.

**Strengths:**
1. The research addresses a high-impact and well-motivated problem in the field of AI in Education by focusing on realistic student simulation, contributing meaningfully to LLM-based math education.
2. The framework introduces a novel cycle-consistent filtering mechanism for model-based quality control of synthetic error examples without extra labels.
3. The framework trains both forward (misconception to answer) and inverse (answer to misconception) models. This is a key difference from STaR, which only trains a forward reasoning model and requires target labels in order to do so. It also mirrors teachers' dual roles, strengthening its AI in Education contribution.

**Weaknesses:**
1. The evaluation lacks sufficient depth and breadth, specifically regarding the choice of models and the absence of human or expert validation. The reliance on a single dataset and a limited set of LLM families makes it difficult to determine if the findings generalize across different educational domains or more modern model architectures. There is also an absence of comparison against established baselines in knowledge tracing and a lack of analysis regarding where cycle consistency might fail.
2. Reviewers expressed significant concern regarding the overall performance and practical value of the framework, noting that the fine-tuned Llama model still lags behind baselines like GPT-3.5.
3. The paper does not provide enough educational insight or theoretical justification to prove that the generated errors reflect genuine human cognitive processes rather than internal LLM logic.
4. The claim of being "unsupervised" is misleading as it uses cycle consistency and correct answer signals, requiring terminology correction.

**Reviewer Concerns:**

During the rebuttal, the authors incorporated new experiments using Qwen3-8B to address concerns regarding limited model family coverage, noting that while preliminary results are available, the process is still ongoing; furthermore, regarding the perceived missing or insufficient baselines, the authors clarified that the requested baselines are already included in the paper. Additionally, the authors revised the terminology from unsupervised to self-supervised, updated the introduction to better ground the tasks in real-world educational use cases, and clarified within the manuscript that the primary goal of the work is not to generate rationales that are themselves human-like.

**Reviewer Scores:**

Given that multiple reviewers have questioned the limited coverage of model families, the experiments involving Qwen3-8B remain preliminary, and concerns regarding the baselines being too weak or incomplete—as well as the lack of real human or teacher evaluation—have not yet been addressed,  reviewers might raise their scores but likely not by a significant margin. A new round of revise and resubmit may be necessary.

---

### Decision · Program_Chairs · 2026-01-26

Reject